# Development of Solid Lipid Nanoparticles by Cold Dilution of Microemulsions: Curcumin Loading, Preliminary In Vitro Studies, and Biodistribution

**DOI:** 10.3390/nano9020230

**Published:** 2019-02-08

**Authors:** Daniela Chirio, Elena Peira, Chiara Dianzani, Elisabetta Muntoni, Casimiro Luca Gigliotti, Benedetta Ferrara, Simona Sapino, Giulia Chindamo, Marina Gallarate

**Affiliations:** 1Dipartimento di Scienza e Tecnologia del Farmaco, Università degli Studi di Torino, 10125 Torino, Italy; daniela.chirio@unito.it (D.C.); chiara.dianzani@unito.it (C.D.); elisabetta.muntoni@unito.it (E.M.); benedetta.ferrara@unito.it (B.F.); simona.sapino@unito.it (S.S.); giulia.chindamo@edu.unito.it (G.C.); marina.gallarate@unito.it (M.G.); 2Interdisciplinary Research Center of Autoimmune Diseases, Department of Health Sciences, “A. Avogadro” University of Eastern Piedmont, 28100 Novara, Italy; luca.gigliotti@med.unipmn.it

**Keywords:** poorly water-soluble drug, nanoparticles, microemulsions, cell line, cancer chemoprevention, drug delivery system

## Abstract

**Background**: Solid lipid nanoparticles (SLNs) are attractive drug delivery systems for lipophilic molecules like curcumin (CURC) with low chemical stability. **Methods**: A simple, innovative, and cold-operating method, named “cold dilution of microemulsion” is developed by the authors to produce SLNs. An oil-in-water microemulsion (µE), whose disperse phase consisted of a solution of trilaurin in a partially water-miscible solvent, was prepared after mutually saturating solvent and water. Trilaurin SLNs precipitated following solvent removal upon water dilution of the µE. After SLN characterization (mean size, Zeta potential, CURC entrapment efficiency, and over time stability), they were tested for in vitro cytotoxicity studies on pancreatic adenocarcinoma cell lines and for in vivo preliminary biodistribution studies in Wistar healthy rats. **Results**: CURC loaded SLNs (SLN-CURC) had mean diameters around 200 nm, were negatively charged, stable over time, and able to entrap CURC up to almost 90%, consequently improving its stability. SLN-CURC inhibited in vitro pancreatic carcinoma cell growth in concentration-dependent manner. Their in vivo intravenous administration suggested a possible long circulation. **Conclusions**: These results, according to a concomitant study on chitosan-coated SLNs, confirm the possibility to apply the developed SLN-based delivery systems as a means to entrap CURC, to improve both its water dispersibility and chemical stability, facilitating its application in therapy.

## 1. Introduction

Curcumin (CURC) is a polyphenolic yellow pigment derived from the rhizome of *Curcuma longa*, which has been and is currently subjected to intensive investigations because of its multiple therapeutic properties.

CURC has several well-documented therapeutic effects [1], including antidiabetic, antihypertensive, anticancer, anti-inflammatory, and antimicrobial properties. Owing to its multitargeting ability in several pathological conditions, it has long been considered as a potential therapeutic agent. Moreover, CURC is described as having a role in reversing multidrug resistance (MDR) through the modulation of Pg-p, inhibiting its function in several in vitro cell line and in vivo tumor models [2].

Unfortunately, like many other lipophilic small molecules, CURC’s efficient use in clinical scenarios is limited by its low hydrophilicity and poor intrinsic dissolution rate, by unfair physical and chemical stability, low bioavailability due to the pitfalls of poor aqueous solubility [3], poor pharmacokinetics, rapid metabolism, and low penetration and targeting efficacy.

By conjugating CURC to metal oxide nanoparticles or entrapping it in lipid/polymeric nanoparticles, liposomes, dendrimers, nanogels, nanoemulsions [4], and polymeric nanoparticles, its water-solubility and bioavailability can be improved and thus increase its pharmacological effectiveness [5,6]. Some Authors [7] developed chemically crosslinked hydrogels containing CURC as hydrophobic drug molecule models and proposed them for prolonged delivery of CURC.

Solid lipid nanoparticles (SLNs) are attracting a number of researchers as an alternative novel drug delivery system for lipophilic molecules like CURC. As assessed by literature data, SLN entrapment not only improves CURC bioavailability [8], but also increases its chemical stability [9]. Additionally, particle size plays an important role in controlling the efficiency of transporting and delivering therapeutic agents by conventional ways, because small sized-particles are accompanied by high surface area, which is particularly suitable to modulate drug delivery to the target organ [10].

Moreover, SLNs represent an alternative option to the traditional colloidal drug delivery systems, such as emulsions, liposomes, and polymeric micro and nanoparticles, because the problems related to industrial production scale-up, sterilization, and mid-term storage are reduced [11]. SLNs are composed of a solid lipid matrix often covered with a surfactant or phospholipid layer; indeed, their main advantage in drug delivery is related to their nanometric size (below 1000 nm) and to the presence of physiological or biocompatible lipids or lipid molecules, solid at room conditions, with a history of safe use in therapy. Lipophilic, as well as hydrophilic drugs (thanks to hydrophobic ion pairing or lipophilic pro-drug preparation), can be loaded in SLNs with high entrapment efficiency.

Over the last 20 years, several methods of SLN production have been thoroughly described in literature: high-pressure homogenization [12], hot microemulsions [13], solvent-based methods such as solvent injection [14], solvent diffusion or evaporation from oil in water (O/W) emulsion [15], and fatty acid coacervation [16].

Most of these techniques require critical process parameters, like high temperatures and/or high pressures that may cause both thermodynamic and mechanic stress to the resulting SLNs; moreover, solvents used are neither completely lacking of toxicity, nor physiologically compatible with the human body. Fatty acid coacervation is a rather simple preparation method, which avoids the use of solvents and operates at rather mild temperatures; indeed, its application is limited only to the use of fatty acid as SLN lipid matrix. In addition, the sudden pH variation (from very alkaline to acidic conditions) can hinder the entrapment of drugs whose stability is negatively influenced by acidic or alkaline conditions.

In this experimental paper, a new method of SLN preparation called “cold dilution of microemulsion” [17] is described. It can be considered as an evolution of the "solvent diffusion" method described in the literature [15]. This production method involves the preparation of an O/W µE at room temperature, whose hydrophilic and lipophilic phases are mutually saturated, using a solid lipid dissolved in a partially water soluble organic solvent as oil phase. SLNs precipitate as a lipid solid matrix following solvent removing upon µE water dilution. Such a method combines the advantages of the emulsion solvent diffusion technique with the high stability and the super solvent properties of microemulsion systems. This technique does not require low or high temperatures, other than the method patented by Gasco [13] which exploited a molten lipid substance added of a hot aqueous surfactant-cosurfactant dispersion. Moreover, in our method neither pH modification, nor ultrasonication, nor homogenization, nor pressure variations are required, nor are toxic solvents used in the view of a future industrial production and therapeutic application.

Finally, the possibility of loading CURC in SLNs prepared by the “cold dilution of microemulsion” technique is exploited to study in vitro their possible application as anticancer agent on different pancreatic tumor cell lines, and to preliminary test their biodistribution patterns in laboratory animals.

## 2. Materials and Methods

### 2.1. Chemicals

Trilaurin (TL), tristearin, myristic acid, glyceryl monostearate, ethyl acetate (EA), benzyl alcohol (BenzOH), triacetin (TA), butyl lactate (BL), polysorbate (20, 40, 80), sodium taurodeoxycholate (Na TdC), sodium taurocholate (Na TC), sodium glycocholate (Na GC), sodium cholate (Na C), Pluronic^®^ F68, polyvinyl alcohol (PVA) 9000, PVA 14000, 1,2 propanediol, curcumin and Sepharose^®^ CL 4B were purchased from Sigma (Dorset, UK). Trimyristin was from Alfa Aesar (Karlsruhe, Germany), glyceryl dibehenate was from Gattefossé (Weil am Rhein, Germany), Epikuron^®^ 200 (lecithin-phosphatydil coline 92%) from Cargill (Minneapolis, MN, USA), and Cremophor^®^ RH 60 (PEG-60 hydrogenated castor oil) from BASF (Ludwigshafen, Germany).

### 2.2. SLN Preparation

SLNs were prepared by the method named “cold dilution of microemulsion” [17]; briefly, a preliminary screening on several compositions was performed, operating at room temperature, to obtain the suitable O/W µE whose disperse phase consisted in a solution of a solid lipid dissolved in a partially water miscible solvent. The solvent and the external phase, consisting of water, were mutually saturated at 25 ± 2 °C for 2 h in order to ensure the initial thermodynamic equilibrium of both liquids, before using them in µE formulation. Lipid nanoparticles were precipitated by quickly adding 5 mL of water into the µE (1 mL) to remove the solvent from the disperse phase and extract it into the continuous phase.

µEs were obtained with biocompatible GRAS ingredients (Generally Recognized As Safe). Among different lipids tested (TL, trimyristin, tristearin, myristic acid, glyceryl dibehenate, and glyceryl monostearate), a TL solution in EA was chosen as oil phase, because of its highest solubility in this partially water-miscible solvent (Table 1). EA and water were mutually pre-saturated before using them in µE preparation (EA_s_ and W_s_ respectively). BenzOH, TA, BL were also tested as partially water-miscible solvents (water saturated) to solubilize the lipid.

Epikuron^®^ 200 was chosen as surfactant together with polysorbate (20, 40, 80) or Cremophor^®^ RH 60 at 3:1 *w/w* constant ratio.

Na TdC, Na TC, Na GC, Na C were tested as co-surfactants, BenzOH was chosen as a co-solvent.

A formulation study was performed varying the percentages of surfactant and co-surfactant/co-solvent. The optimal µE formulation, in the absence of any drug, called µE1, is reported in Table 2.

µE1 (1 mL) was then diluted with a 2% *w/w* polymeric aqueous solution (5 mL) to precipitate SLNs. In order to avoid SLN aggregation [16], different polymers (Cremophor^®^ RH60, Pluronic^®^ F68, PVA^®^ 9000, PVA^®^ 14000) and different percentages of Pluronic^®^ F68 were tested to check the best conditions to obtain small and non-aggregated SLNs. Probably, the polymer disposition on SLN surface influences surface hydrophilicity and charge. A formulation study was then performed to optimize SLN size.

### 2.3. Pseudo-Ternary Diagrams

Phase diagram was constructed at room temperature. The resultant phase behavior was mapped on pseudo-ternary phase diagrams, constructed by titrating a series of lipid and surfactant/co-surfactant mixtures with aqueous phase (W_s_/BenzOH mixture at constant 13:1 *w/w* ratio). Epikuron^®^ 200-Cremophor^®^ RH60-Na TC (3:1:0.6 *w/w/w*) was used as surfactant/co-surfactant mixture, while the lipophilic phase consisted of a TL solution (300 mg/mL) in EA_s_. Appropriate amounts of surfactant/co-surfactant and lipophilic phases were weighted (1 g) into glass ampules, shaken for sufficient time to attain equilibrium, and then progressively enriched with the aqueous phase, added dropwise. The amounts of aqueous phase at which the transparent/opaque transition occurred were used to determine the phase domains. The domain of transparent, isotropic systems with mean diameter >5 nm was considered as the µE phase, while the domain of existence of turbid systems was classified as the emulsion phase. The domain of liquid crystals was determined by interposing the system between two polarized lenses and observing the different optical properties of µE and liquid crystalline systems [18].

For the purposes of this study, a schematic representation was sufficient as a guide to follow the evolution of phase equilibrium. This experimental procedure was repeated for other points of surfactant/co-surfactant to lipid weight ratios, and the corresponding phase diagram was constructed.

### 2.4. CURC Loaded SLNs

With the aim to check the potential applications of µE-derived SLNs in anticancer drug delivery, CURC was incorporated in the µE whose composition is described in Table 2 (µE2), in which Na TC was the selected bile salt. To obtain the maximum achievable CURC loading without negatively affecting the stability of the system, it was necessary to substitute BenzOH with 1,2 propanediol. In such a way, increasing amounts of drug, up to 11 mg/mL µE, were introduced in the formulation described in Table 2, in order to obtain the maximum loading in the lipid matrix. µE2 (1 mL) was diluted with a 2% *w/w* Pluronic^®^ F68 aqueous solution (5 mL) to precipitate CURC loaded SLNs.

### 2.5. Gel Filtration of SLNs

CURC loaded SLNs and free CURC were separated by gel filtration. A volume of SLN suspension (1 mL) was put on the top of the gel filtration (GF) column and the sample was then eluted by gravity, adding a hypertonic phosphate buffered saline (NaCl and KCl at 2:1 ratio). The stationary phase was a matrix of cross-linked agarose (Sepharose^®^ CL 4B). Opalescent fractions containing purified SLNs, whose scattering was monitored by DLS technique (90 Plus-Particle Size Analyzer, Brookhaven Instruments Corporation, Long Island, NY, USA), were pooled and concentrated under N_2_ or by freeze-drying, without adding any cryoprotectant using a Modulyo Freeze Dryer (Edwards Alto Vuoto, Trezzano sul Naviglio, Italy). When the volume of the N_2_-flushed suspension was reduced to almost 1 mL, N_2_ flux was stopped and SLNs were analyzed for particle size, shape, and CURC concentration.

Freeze-dried SLNs were reconstituted in 1 mL water and they were analyzed for particle size determination.

Gel filtration should eliminate drug or surfactants/co-surfactants excess adsorbed to SLN surface.

### 2.6. Particle Size and Zeta Potential Determination

Size distribution, polydispersity index (PI), and Zeta potential measurements were determined 1 h after SLN preparation using DLS technique. Size measurements were obtained at an angle of 90° at 25 °C. Before the analysis, SLN suspensions were 1:20 diluted with MilliQ water for size determination or with 0.01 M KCl for Zeta potential determination, in order to achieve the suitable conductivity. Size measurements were also recorded diluting samples with the grown medium (DMEM) used to culture cells to mimic the conditions under which SLNs undergo in vitro experiments. All data were calculated on triplicate samples.

### 2.7. Differential Scanning Calorimetry (DSC) 

DSC studies were performed to investigate CURC-lipid interactions and the crystallinity of both CURC and lipid, because these parameters might influence the release.

Perkin Elmer differential calorimeter (DSC7, Perkin Elmer, Nortwalk, CT, USA) equipped with an instrument controller Tac 7/DX (Perkin Elmer) was used. A heating rate of 10 °C/min was employed in the 25–200 °C temperature range. SLN suspensions were freeze-dried without adding any cryoprotectant using a Modulyo Freeze Dryer (Edwards Alto Vuoto, Trezzano sul Naviglio, Italy). Freeze-dried SLNs were weighted and placed in a conventional aluminum pan for analysis.

The degree of crystallinity of SLNs was estimated by calculating the ratio between the melting enthalpy/g lipid in SLN dispersion and the melting enthalpy/g of the bulk material [19,20].

### 2.8. Scanning Electron Microscopy (SEM)

Surface morphology of SLNs was obtained by SEM using Stereoscan 410 (Leica, Wetzlar, Germany). The nanoparticle suspension was fixed on a metallic stab, evaporated under vacuum and then coated with gold (SCD 050, Leica, Wetzlar, Germany) under vacuum. The gold-coated particle layer was scanned using the accelerating voltage scanning of 20 kV.

### 2.9. In vitro Quantification of CURC 

High-performance liquid chromatography (HPLC) analysis was performed using a LC9 pump (Shimadzu, Tokyo, Japan) with an Inertsil^®^ ODS-2 5 μm 150 × 4.6 mm column and a C-R5A integrator (Shimadzu, Tokyo, Japan); detector: UV-Vis λ = 450 nm (Shimadzu, Tokyo, Japan).

CURC was eluted in isocratic conditions at a flow rate of 1.0 mL/min using methanol/acetic acid 3.5% (70:30 *v/v*) as mobile phase. Injection volume was 20 μL, and retention time of CURC was 6 min.

A calibration curve with acceptable linearity (*R*^2^ = 0.9998) was constructed by plotting the peak area versus CURC concentration within 0.2–20 μg/mL concentration range. The relative standard deviation (RSD) of intra- and inter-day precision at three concentrations (0.2, 2, and 20 μg/mL) was less than 3%, and the accuracy ranged from 97.0% to 102.5%.

### 2.10. CURC Concentration in SLN Suspension and Entrapment Efficiency (%EE)

CURC concentration in SLN suspension after gel filtration (GF) was determined by the HPLC method used for in vitro quantification of CURC. The drug amount resulting after GF of SLNs was considered as totally entrapped in the SLN matrix. %EE is calculated as the ratio between the amount of SLN-entrapped drug (post-column drug concentration) and the total amount used in SLN preparation (pre-column drug concentration) × 100.

### 2.11. CURC Release from SLNs

In vitro CURC release was determined using the non-equilibrium dialysis method [21]. A multicompartmental rotating cell system was used; donor and receptor compartments had a volume of 1.5 mL each. The experiments were carried out at room temperature. As hydrophilic membrane, Servapor^®^ dialysis tubing (Serva, Heidelberg, Germany), cut-off 12,000–14,000 Da, was used; the effective membrane area was 2 cm^2^.

The receiving medium was 10% *v/v* PEG 300 aqueous solution. PEG 300 was added to improve CURC dissolution in an aqueous medium, maintaining sink conditions. A saturated CURC reference solution in 10% *v/v* PEG 300 and a CURC-loaded SLN suspension, obtained from µE2 containing 9 mg/mL CURC, and called SLN-CURC, were studied as donor formulations. At fixed times, the receptor solution was tipped out and the cell was refilled with fresh receiving medium. Drug concentration in the receptor phase was determined by the HPLC method for in vitro quantification of CURC. The results were evaluated as CURC apparent permeability constant, *Kp*_app_ (cm h^−1^) calculated from the slope of the straight section of the curve obtained by plotting the amount of CURC diffused from the donor formulation versus time, assuming pseudo zero-order kinetics.

### 2.12. SLN and CURC Stability

SLN mean diameters and CURC concentration in SLN-CURC, stored at 4 ± 1 °C, were monitored for 30 days at fixed time intervals to study SLN physical stability and CURC chemical stability in SLNs over time. Mean sizes and Zeta potential were determined by DLS, whilst CURC concentration (µg/mL) in SLN dispersion after GF was determined by HPLC, as described above.

As pure CURC is highly unstable to chemical degradation in alkaline aqueous solutions (pH ≥ 7.0) [22], CURC chemical stability in SLN-CURC was also evaluated after incubating them in phosphate buffer solution (PBS) at pH 7.4 for 24 h and comparing it to a saturated solution of CURC in the same phosphate buffer. For this purpose, native CURC was dissolved in PBS pH = 7.4 with the help of PEG 300 (final PEG concentration 10% *v/v*). CURC was added up to saturation (15 µg/mL).

### 2.13. Empty SLN in vitro Toxicity Studies on Podocytes and HUVEC

In this study, lines of immortalized human podocytes obtained by infection of cultures of renal cells with a hybrid Adeno5/SV40 virus were used. Primary cultures of human podocytes were kindly provided by Prof. G. Camussi (Department of Internal Medicine and Center for Molecular Biotechnology, University of Turin, Turin, Italy).

Podocytes were cultured in Dulbecco’s modified Eagle’s medium (DMEM, Manufacturer, city, country) supplemented with fetal bovine serum (FBS) 10%, penicillin (100 IU·mL^−1^), streptomycin (100 µg·mL^−1^), and l-glutamine (2 mM).

HUVEC were isolated from human umbilical veins by trypsin treatment (1%) and cultured in M199 medium with the addition of 20% fetal calf serum (FCS), 100 U/mL penicillin, 100 μg/mL streptomycin, 5 UI/mL heparin, 12 μg/mL bovine brain extract, and 200 mM glutamine. HUVEC were grown to confluence in flasks and used at the 2nd–5th passage. The use of HUVEC was approved by the Ethics Committee of the “Presidio Ospedaliero Martini” of Turin and conducted in accordance with the Declaration of Helsinki. Written informed consent was obtained from all donors.

The day before the experiment, podocytes and HUVEC were plated on 96-well culture plates (3 × 10^3^ cells per well). Then, they were treated with empty SLNs (1% *w/w* TL), progressively diluted, and after 72 h incubation, the amount of viable cells was evaluated by MTT (Sigma-Aldrich, Milano, Italy) inner salt reagent at 570 nm, as described by the manufacturer’s protocol. The controls (i.e., cells that had received no drug) were normalized to 100%, and the readings from treated cells were expressed as viability inhibition percentage.

### 2.14. SLN-CURC in vitro Cytotoxicity Studies

The human tumor cells CFPAC-1 and PANC-1 were obtained from ATCC (Milan, Italy) and used to perform in vitro cytotoxicity test. CFPAC-1 and PANC-1 cell lines (800/well) were seeded in 96-well plates and incubated at 37 °C, 5% CO_2_ humidified atmosphere for 24 h in DMEM supplemented with FBS 10%, penicillin (100 IU·mL^−1^) /streptomycin (100 μg·mL^−1^), and l-glutamine (2 mM).

Then, they were treated with free CURC, SLN-CURC diluted to obtain CURC in the 0.5–10 µM concentration range, based on CURC incorporation and empty SLNs with the same dilution. After 72 h incubation, the amount of viable cells was evaluated by MTT. The controls (i.e., cells that had received no drug) were normalized to 100%, and the readings from treated cells were expressed as viability inhibition percentage.

### 2.15. Biodistribution of SLNs 

Because the bioavailability of orally administered CURC is poor, primarily due to low absorption, rapid metabolism, and fast elimination, in our study we opted to investigate the parenteral administration of CURC.

SLN-CURC and CURC solution in normal saline with 10% *w/w* Tween^®^ 80 (surfactant allowed for intravenous administration, here used to increase CURC solubility) were administered at 2 mg/kg dose through a catheter surgically positioned in the jugular vein of male Wistar healthy rats (weight 250 g, age 3 months). Each experiment was performed on 4 rats for each administered sample.

The procedures conformed to the Ethics Committee of University of Turin’s institutional guidelines on animal welfare (D. Lgs. 26/2014) as well as international Guidelines, and all efforts were done to minimize the number of animals and their discomfort (3R guidelines). All experiments on animal models were performed according to an experimental protocol approved by the University Bioethical Committee, and authorized of Italian Ministry of Health (authorization n. 0165/2015).

At scheduled times (30, 60, and 120 min after administration), rats were sacrificed by CO_2_-induced euthanasia; plasma was withdrawn, and organs (liver, spleen, kidneys, lungs, heart, pancreas, and brain) were removed surgically. Organs were homogenized with UltraTurrax^®^ (IKA, Staufen, Germany) for 5 min in water at a tissue concentration of 250 mg/mL. Tissue homogenates and plasma were treated with CH_3_CN (1:5 *v/v*) as protein precipitant, vortexed for 60 s and centrifuged at 6000 rpm for 5 min. The organic phase was collected and analyzed by HPLC method performed through a YL9100 HPLC system equipped with a YL9110 quaternary pump, a YL9101 vacuum degasser and a Shimadzu RF-10A fluorescence detector (Shimadzu, Tokyo, Japan), linked to YL-Clarity software for data analysis (Young Lin, Hogye-dong, Anyang, Korea).

HPLC analysis was performed with LiChrospher^®^ RP18 5 μm 250 × 4.6 mm column, flow rate = 1 mL/min; A = 3.5% *v/v* acetic acid in water, B = methanol, gradient (B%, time (min): 40, 10; 90, 5; 40, 1): *Rt* = 12 min (λ_ex_ = 409 nm, λ_em_ = 535 nm).

### 2.16. Data Analysis

Data are shown as mean ± S.E.M. Statistical analyses were performed with Prism 3.0 software (GraphPad Software, La Jolla, CA, USA) using one-way ANOVA and the Dunnett test.

## 3. Results and Discussion 

### 3.1. SLN Formulation and Characterization

TL dissolved in EA_s_ was chosen as lipid matrix, as EA is the only solvent, among those taken into account, able to solubilize TL, allowing to obtain a sufficiently concentrated lipid solution.

When 300 mg/mL TL was dissolved in water-saturated solvent (EA_s_), its resulting percentage in µE was almost 3%, considering that 200 µL of such solution were used to obtain the µE.

After identifying the quali-quantitative composition of the µE, the pseudo-ternary phase diagram was constructed (Figure 1). The top apex of the system represents the lipophilic phase (EA_s_-dissolved TL) and the remaining two apices represent the surfactant/co-surfactant mixing ratio (lecithin-PEG-6 hydrogenated castor oil-Na TC) and aqueous phase (W_s_/BenzOH). The region M on phase diagram indicates the fluid transparent isotropic µE domain, in which µEs were stable for a sufficiently long time (at least 1 month) at room temperature (25 °C). Birefringent liquid crystalline regions are marked LC. Unmarked areas indicate multiphase regions.

The construction of the pseudo-ternary diagram was necessary to identify the boundaries between the areas of the M and LC existence fields, since only by diluting microemulsion system was it possible to obtain SLN precipitation. No difference between µE with 1,2 propanediol or BenzOH was noted about the boundaries of existence field of M and LC.

The formulation without CURC (µE1) with Na TC as bile salt, reported in Table 2, was chosen to precipitate SLNs in water in the presence of different polymers used as stabilizers. After dilution, the partially miscible solvent (EA_s_), present in the oil phase, diffused into the un-saturated aqueous solution determining the precipitation of spherical-shaped TL-SLN, which were in the submicron size range. It is very difficult to foresee if SLN mean diameters will keep the size of oil nanodroplets originally present in the µE. Indeed, even if µE droplets are generally comprised in 10–150 nm, it is very hard to measure them directly in the µE by DLS or by microscopy without perturbing (by diluting or pretreating) the equilibria of a liquid system. Moreover, it might be that SLN mean diameters are higher than those of µE droplets, as in the precipitation step, as certain droplet aggregation can occur.

The selected SLNs, consisting of a TL core, were stabilized with different polymer solutions.

Formulation studies were carried out screening different polymers to select the one allowing the formation of stable SLN suspension, avoiding nanoparticle aggregation. In Table 3, mean diameter and PI of empty SLNs, obtained from µE1 after precipitation of the lipid matrix in 2% *w/w* aqueous solution of different polymers, are reported. The smallest SLN diameter was obtained using Pluronic^®^ F68 as stabilizer. Pluronic^®^ F68, unlike PVAs and Cremophor^®^ RH60, is an amphiphilic polymer, whose lipophilic probably strongly interacts with the lipophilic surface of SLNs. 

A percentage of Pluronic^®^ F68 ranging from 2 to 3% *w/w* produced SLNs with the smallest diameters (Figure 2). Probably, a percentage of 1% *w/w* was insufficient to stabilize SLN surface, whereas 4% *w/w* might adsorb onto SLN surface to a greater extent, determining a sharp increase in mean diameters. Then 2% *w/w* Pluronic^®^ F68 aqueous solution was chosen to dilute the subsequent microemulsion formulations.

A further modification of µE formulation was performed screening different bile salts in order to select the best performing one regarding SLN mean diameters and PI. Indeed, bile salts are naturally occurring substances whose amphipathic, detergent-like characteristics determine a wide range of biological activity; the ability to form micellar aggregates both in pure and in mixed form with Epikuron^®^ 200 was already exploited by employing them as co-surfactants in lecithin-stabilized µE [23].

Several SLNs were obtained using Na TC, Na TdC, Na GC or Na C as co-surfactant, whose mean diameters are reported in Table 4: among the different bile salts screened, Na TC allowed to obtain SLN with the smallest size (152 nm). Perhaps the deoxidation at the 7 position of Na TdC is responsible for its higher rigidity and hydrophobicity than Na TC, which can hinder the interaction with the lipid matrix; moreover, in the cholate series, the presence of taurine confers suitable amphiphatic properties to the bile salt.

For this reason, Na TC-containing µE1 was selected for further experiments.

When CURC was added to µE1, some adjustments were needed in its formulation to maintain transparency and stability, which are necessary conditions of µEs: BenzOH was replaced by 1,2 propanediol up to 15.10% *w/w*, obtaining µE2, in which it was possible to load increasing amounts of CURC (up to 11 mg/mL). 

Both empty and CURC-loaded SLNs, obtained by diluting µE in aqueous Pluronic^®^ F68 solution, were purified by GF technique. 

CURC loaded SLN size slightly increased upon increasing amounts of loaded CURC, compared to their respective empty SLNs; this is probably due to the presence of CURC inside of the lipid core; however, mean diameters of SLNs formed from µE2 did not markedly exceed 200 nm (Table 5). All formulations were negatively-charged: Zeta potential varied from −10 to −19 mV upon increasing amount of loaded CURC, indicating a relatively fair stability and dispersion of the systems. Generally, the presence of both Epikuron^®^ 200 and Na TC in SLN formulations determines a marked negative charge onto nanoparticle surface; in this case, the slightly negative value of Zeta potential was attributed to the presence of Pluronic^®^ F68, a non-ionic surfactant used in the production of relatively stable dispersions. As it is non-ionic, the surfactant can shield the electric double layer. Pluronic^®^ F68 can also provide additional steric stabilization of particles [24,25].

CURC %EE slightly decreased, whilst, CURC loading in SLNs increased as a function of increasing amounts of CURC in µE up to 9 mg CURC; no further increase in CURC loading was noted for higher CURC amounts. A mass of 9 mg CURC corresponded to a 112.5 µg CURC/mg lipid loading, which can be considered as the saturation concentration of CURC in SLN lipid matrix.

SLNs obtained from µE2 containing 9 mg/mL CURC, called SLN-CURC, were used for further experiments.

The satisfactory CURC amounts recovered in SLN dispersion indicate the efficiency of the new preparation method of SLNs from µE, avoiding significant CURC losses during formulation. 

After GF, SLNs had to be reconstituted to the initial volume by freeze-drying followed by further water addition up to the initial volume, or by direct concentration to the initial volume under N_2_ flux. The concentration under N_2_ flux did not determine any remarkable mean diameter increase, meanwhile after freeze-drying some SLN aggregation occurred (Table 6). Growth media do not seem to influence SLN dimensions, as all formulations maintained their size (± 5%).

The surface morphologies of SLNs were studied using SEM, which confirmed the shape of SLNs, which were well-dispersed and separated from each other, and looked to be single unities, suggesting the lack of aggregates (Figure 3).

To meet the requirement of using SLNs as drug-carrier systems, their solid state after µE dilution had to be verified. The status of lipid particles was investigated using differential scanning calorimetry (DSC); moreover, DSC analysis was also aimed to investigate the degree of lipid crystallinity in SLNs and the eventual presence of polymorphs, as these parameters are strongly correlated with drug incorporation and release profiles [20].

DSC thermogram of TL (Appendix A), a monoacid triglyceride containing a 12C-chain for each fatty acid esterified with the glycerol molecule, exhibits a sharp endotherm peak, with onset at 49 °C, which indicates the melting of the stable β-form of TL [26]. 

The thermogram of SLN-CURC showed a broadened endotherm peak with onset at 34 °C, probably due to the melting of the unstable β′-form, and no supercooled melt was revealed. As further hypothesis, according to Siekmann and Westesen [19], the melting point decrease of SLN systems might be due to the colloidal nature of the particles, in particular to their high surface-to-volume ratio and not the recrystallization of the lipid matrices in a metastable polymorph. DSC analysis of CURC showed the melting point at 173.17 °C. Moreover, it is interesting to note that it was not possible to observe the transition peak of CURC in SLN thermograms, confirming CURC entrapment in SLNs.

CURC melting peak was not recorded in the SLN formulation, in accordance with the location of the drug within the solid lipid matrix.

A degree of crystallinity higher than 30% was obtained for SLNs: when crystallinity decreased from 100%, a less ordered modification of the lipid matrix is present in SLNs, which might promote drug release [27].

Over-time stability of CURC in SLN-CURC was studied during 30 day-storage. Physical stability was checked by monitoring particle mean size and Zeta potential over time by DLS measurements. No significant modification was noted, either in particle size or in Zeta potential (Figure 4). Furthermore, polydispersity index was stable over time (data not shown).

Also, CURC concentration in SLN suspensions decreases by approximately 3% during 30 day-storage remaining quite unmodified: this result is actually significant, as CURC entrapment in SLNs allowed, if not to hinder, at least to slow down CURC degradation.

Indeed, CURC is a hydrophobic polyphenol compound that is unstable in aqueous solution: in a recent study, Kharat and al. [22] demonstrated the chemical instability of CURC in aqueous solutions at pH ≥ 7.0 and the tendency to crystallize out of aqueous solutions at pH < 7, the latter phenomenon being particularly relevant at low temperatures. 

To confirm whether our formulation can enhance CURC stability in aqueous media, we incubated SLN-CURC in PBS (0.1 M, pH 7.4) for 24 h. Monitoring CURC concentration in SLN dispersion over time, it resulted to be stable, as almost 94% of the initial amount (1042 µg/mL) was found in SLN dispersion after 24-h incubation, compared with free CURC that in PBS degraded quickly, with only 50% of the initial amount remaining. These results confirmed the role of the developed lipid nanoparticulate system to entrap and to protect CURC from aqueous degradation.

In in vitro release experiments, CURC was released more slowly from SLNs (*Kp*_app_ = 0.015 cm h^−1^) than from reference CURC solution (*Kp*_app_ = 0.075 cm h^−1^).

A 1-hour lag time was noted for SLN-CURC, which might be due to the drug adsorbed on the nanoparticles surface (Figure 5). Subsequently, after 6 h the percentage of CURC released was only 12%, indicating a slow release of CURC from the lipid solid matrices, due to the erosion and degradation of the components of nanoparticles [28,29]. After 24 h, the CURC amount released from SLN was 20% compared to 65% from the reference solution.

At room temperature SLNs are in solid state, as confirmed by DSC analysis. Therefore, the mobility of incorporated drugs is reduced, which is a prerequisite for controlled drug release.

### 3.2. In vitro Cytotoxicity Test

Since SLNs should not be toxic for healthy cells, empty SLN biocompatibility on HUVEC was verified. Cells were treated with increasing dilutions of empty SLN suspension (1% *w/w* of TL) for 72 h. After exposure, cell viability was evaluated assessing the number of viable cells by MTT assay, revealing a cell survival rate higher than 80% for all the dilutions tested (1/100, 1/200, 1/500, 1/1000).

Then we compared the ability of free CURC, empty SLNs, and SLN-CURC to inhibit the growth of CFPAC-1 and PANC-1 cell lines. Cells were cultured in the absence and in the presence of titrated amounts of CURC (0.5–10 µM) for 72 h and the amount of viable cells was then assessed by MTT assay. Figure 6 shows that SLN-CURC inhibited the growth of both cell lines. The effect was concentration-dependent with small differences between the two cell lines. Probably, PANC-1 cells internalize CURC better than CFPAC-1 cells, as is reported in all the tested doses; CURC uptake by PANC-1 cell line is well documented in the literature [30].

A significant increase of inhibition effect was observed at each CURC concentration on cells treated with SLN-CURC with respect to the free compound, while empty SLNs did not display any cytotoxicity. It is noteworthy that, when CURC was loaded in SLNs, an almost 10% cell growth inhibition resulted for a 0.5 µM CURC, a concentration at which free CURC was ineffective. Studies reported that drugs exhibited better cytotoxicity profile when incorporated in SLNs compared to free drug because they have high surface areas for more efficient uptake by cells [31]. These results revealed the effectiveness of SLNs to release the intact molecule inside the targeted cells. The mechanism of internalization of SLNs is thoroughly described in literature as occurring through a receptor-mediated endocytosis through clathrin-coated pits [32].

### 3.3. Biodistribution Studies 

Before biodistribution experiments, it was mandatory to inject empty SLNs (1% *w/w* of TL) in rats to test their biocompatibility: actually, after carefully monitoring rats, neither changes in their behavior, nor alteration in vital signs were noted in rats up to 48 h after empty SLN intravenous administration (nanoparticle mean diameter 193 nm).

Moreover, Tween^®^ 80, which was used to improve CURC aqueous solubility, has been widely used as a solvent for pharmacological experiments [33].

The small number of animals involved in these preliminary investigations hinders one from drawing precise conclusions about quantitative all-body distribution, but the recovered data can give us some general information about CURC tracking after intravenous administration of SLN-CURC, which is summarized as follows.

The preliminary results of biodistribution experiments show that CURC can be delivered by SLN (with mean diameter 173 nm) to the different organs at concentration comparable with those obtained by administering CURC in 10% Tween^®^ 80 aqueous solution.

In Figure 7, CURC concentration in each organ is reported for different post-administration times. When free CURC was administered in 10% *w/w* Tween^®^ 80 aqueous solution, a maximum accumulation in liver, lung, and spleen was provided at earlier stages than for SLN, and a progressive decrease of its concentration over time was noted, in almost all the considered organs. On the contrary, at 60 min post administration, SLN-CURC revealed the highest concentration/vs. time in almost all organs and especially in lung, liver, brain, spleen, and pancreas, suggesting a possible long circulation. These results might be related to the delayed release and the stability increase of SLN-loaded CURC assessed in in vitro release studies.

As it can be predicted, 60 and 120 min after administration marked CURC accumulation in RES organs (spleen or liver) was noted when it was loaded in SLNs compared with free CURC solution. Probably, the small amount of Pluronic^®^ F68 adsorbed onto SLN surface was not sufficient to confer them stealth properties. Indeed, the higher CURC amount found in pancreas at time, when it was administered in SLNs, rather than in 10% *w/w* Tween^®^ 80 aqueous solution, was quite surprising and worthy of further investigation.

CURC was considered a potential drug to both inhibit cancer and overcome drug resistance [9]. Recently, chemoprevention has emerged as an effective tool in the fight against various types of cancers, including pancreatic cancer. Of the various chemopreventive agents being studied, CURC has been proven to be effective in chemoprevention of this cancer. Nevertheless, the clinical translation of this agent has been significantly hampered due to its poor oral bioavailability after administration [34]. Engineering a SLN drug delivery system for CURC offers a means of increasing the bioavailability in the plasma and tissues in comparison to the free form, thereby ultimately improving the therapeutic efficacy.

Considering the successful results obtained in in vitro cytotoxicity tests on PANC-1 and CFPAC-1 cultures, the in vivo results might be a fair short point to further investigation also concerning in vivo pancreas tumor models. 

The above mentioned results, although far from being optimal for an in vivo administration, are encouraging and give us many suggestions about the way to modify SLN composition and surface properties both to minimize opsonization and to prolong circulation time. 

## 4. Conclusions

In conclusion, five end points have been successfully reached:the development of a new, solvent free, cold-operating, and simple-to-realize technique, to obtain physiologically compatible SLN;the feasibility of the µE dilution method for CURC loading;the in vitro efficacy of SLN-CURC on two different pancreatic tumor cell lines;the lack of adverse effects of SLNs on rats after intravenous administrationdifferent biodistribution patterns of CURC when in vivo administered in SLNs and prolonged circulation time when compared with aqueous solution.

These results confirm the possibility to apply the developed SLN-based delivery systems as a mean both to entrap CURC and to improve CURC water dispersibility and chemical stability, which would facilitate its application in therapy.

## Figures and Tables

**Figure 1 nanomaterials-09-00230-f001:**
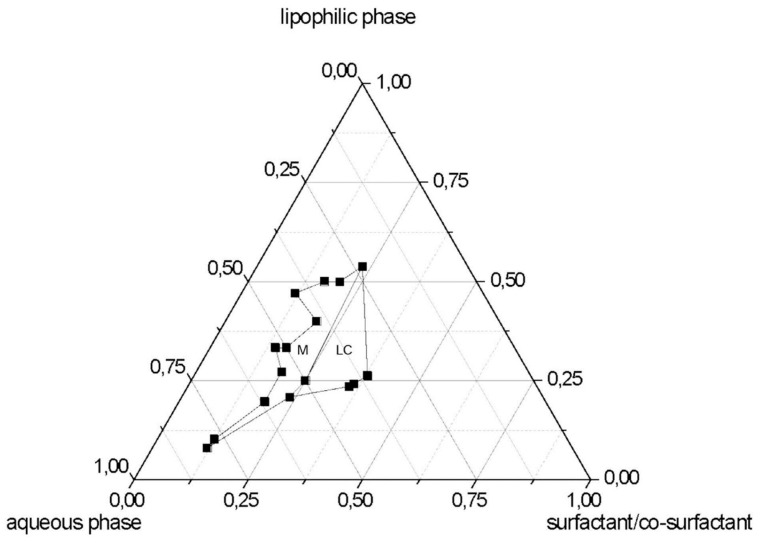
Phase diagram at room temperature (25 °C) of the systems containing lipophilic phase/aqueous phase/surfactant:co-surfactant (trilaurin (TL) solution in EA_s_/W_s_-BenzOH/lecithin-PEG-6 hydrogenated castor oil-sodium taurocholate (Na TC). M: µE domain; LC: liquid crystal domain.

**Figure 2 nanomaterials-09-00230-f002:**
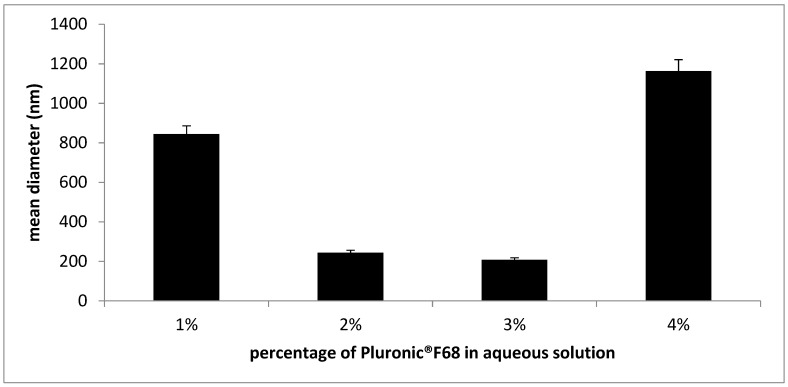
Mean diameter (nm) of empty solid lipid nanoparticles (SLNs) dispersed in Pluronic^®^ F68 aqueous solution at different polymer percentages.

**Figure 3 nanomaterials-09-00230-f003:**
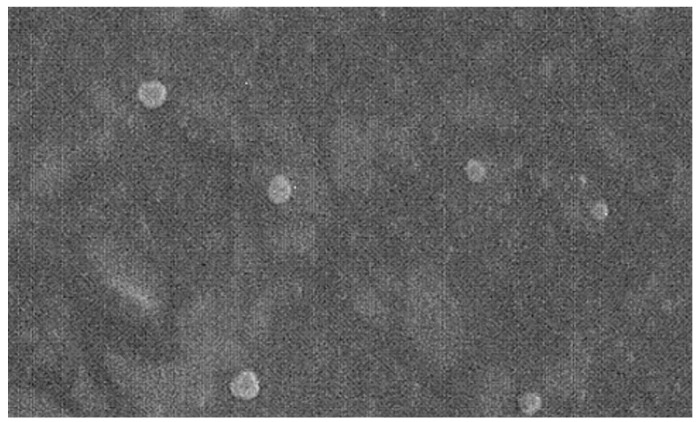
Scanning electronic microscopy (SEM) picture of SLN-CURC (Detector SE1, EHT 15.00 KV, Mag 10.00 KX, 
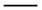
 500 nm).

**Figure 4 nanomaterials-09-00230-f004:**
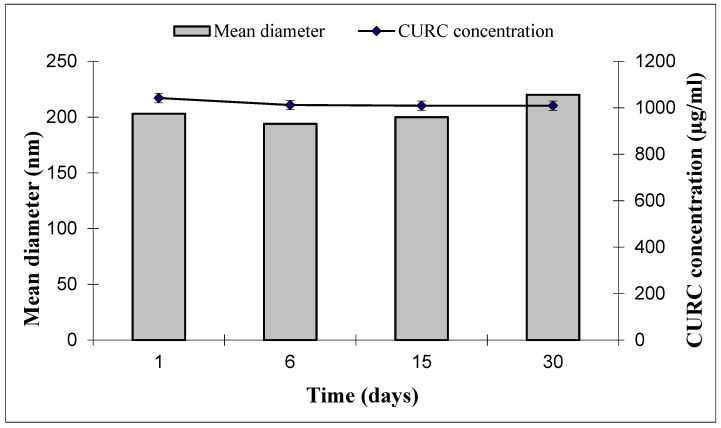
Stability of SLN-CURC in term of mean diameter (nm) and CURC concentration (µg/mL) over time after storage at 4 °C. Zeta potential (mV) was 1 day = −10.02 mV ± 2.32; 6 days = −13.57 mV ± 2.71; 15 days = −18.86 mV ± 2.98; 30 days = −19.53 mV ± 2.77.

**Figure 5 nanomaterials-09-00230-f005:**
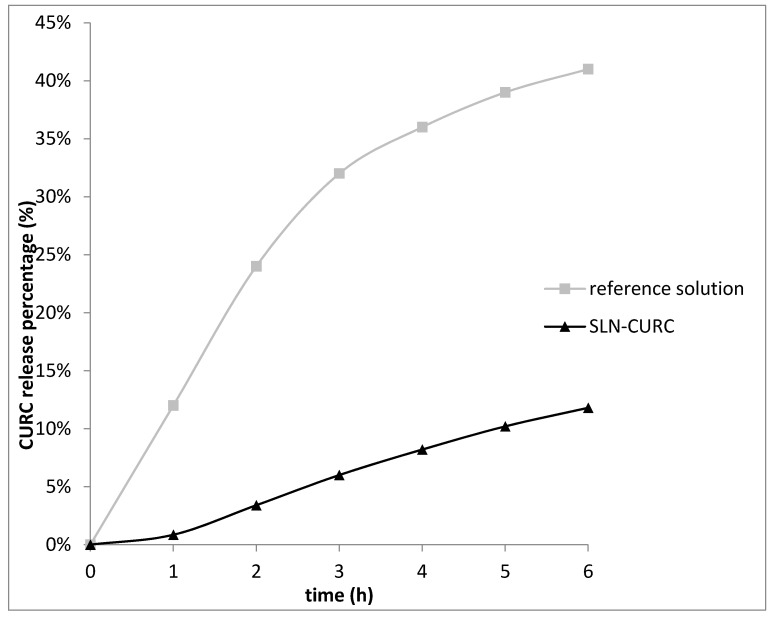
CURC release from SLN-CURC compared to reference solution (up to 6 h).

**Figure 6 nanomaterials-09-00230-f006:**
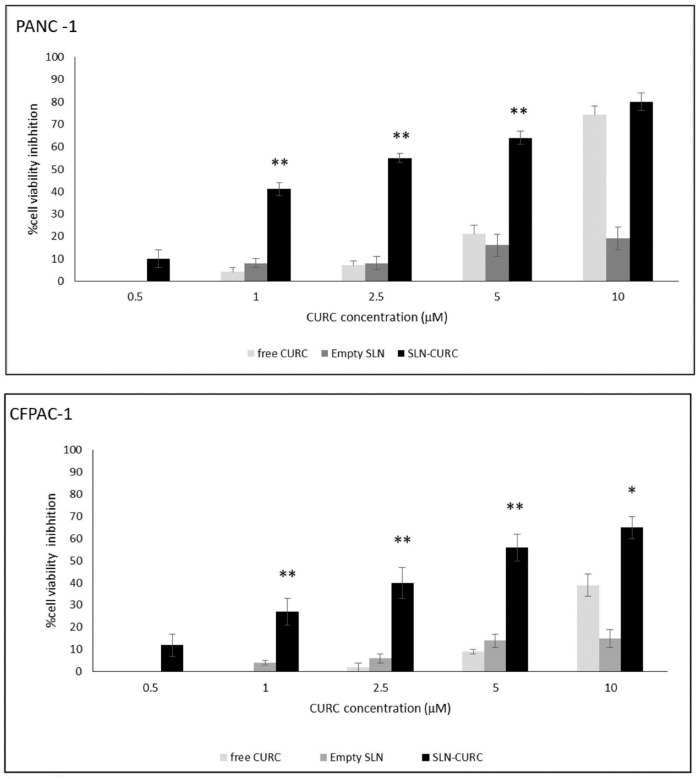
Inhibition of proliferation following empty SLN, SLN-CURC, and free CURC administration. PANC-1 and CFPAC-1 cells (800 cells/well) were treated with increasing concentrations of SLN-CURC and CURC for 72 h; the result was expressed as the percentage of viable cells versus the control expressed as mean ± S.E.M (*n* = 5). Asterisks mean statistically significant differences of SLN-CURC vs. CURC-treated cells at the same concentrations (* *p* < 0.05 and ** *p* < 0.01).

**Figure 7 nanomaterials-09-00230-f007:**
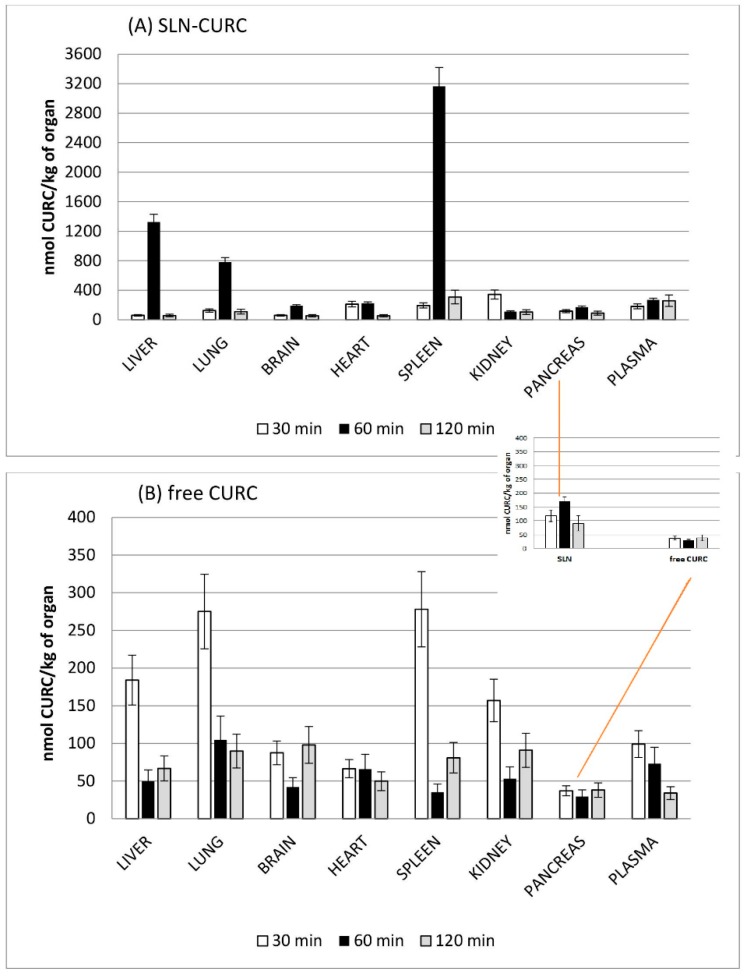
Biodistribution profile over time for each organ of CURC in SLN-CURC (**A**) and free CURC (**B**).

**Table 1 nanomaterials-09-00230-t001:** Solubility (mg/mL) of trilaurin (TL) in water partially miscible solvents and water solubility of the solvents (25 °C).

Solvent	Solubility of TL in Water Saturated Solvent (mg/mL)	Solvent Water Solubility (g/100 mL)
TA	<300	6.1
BL	<300	4.2
EA	>300	8.7
BenzOH	<300	4.3

**Table 2 nanomaterials-09-00230-t002:** µE composition. µE1: empty µE; µE2: µE to be loaded with curcumin (CURC).

Components	µE1 Composition (*w/w* %)	µE2 Composition (*w/w* %)
TL	3.33	4.35
EA_s_	14.97	13.06
Epikuron^®^ 200	12.48	10.88
Cremophor^®^ RH60	4.16	3.63
W_s_	58.22	50.80
Bile salt	2.49	2.18
BenzOH	4.35	–
1,2 Propanediol	–	15.10

**Table 3 nanomaterials-09-00230-t003:** Mean diameter (nm) ± standard error (PI) of empty solid lipid nanoparticles (SLNs) dispersed in different 2% *w/w* polymer solutions.

Samples	Mean Diameter (nm) ± S.E. (PI)
SLNs in PVA^®^ 9000	200.2 ± 2.2 (0.191)
SLNs in Cremophor^®^ RH60	223.9 ± 2.4 (0.224)
SLNs in Pluronic^®^ F68	151.9 ± 3.4 (0.210)
SLNs in PVA^®^ 14000	282.2 ± 11.5 (0.139)

**Table 4 nanomaterials-09-00230-t004:** Mean diameter (nm) ± standard error and pH of empty SLN suspensions in Pluronic^®^ F68, prepared screening different bile salts in µE formulation.

SLN	Mean Diameter (nm) ± S.E.	pH of SLN Suspension
SLNs with Na TdC	332 ± 1.4	4.43
SLNs with Na TC	152 ± 2.4	3.89
SLNs with Na GC	334 ± 1.5	4.57
SLNs with Na C	230 ± 1.7	6.49

**Table 5 nanomaterials-09-00230-t005:** Mean diameter (nm) ± standard error (PI), Zeta potential (mV), and entrapment efficiency (%EE) ± standard error of SLNs with increasing amount of CURC, after purification by gel filtration (GF) technique and concentration under N_2_.

mg of CURC Added	Mean Diameter (nm) ± S.E. (PI)	Zeta Potential (mV) ± S.E.	%EE ± S.E.
3 mg in SLN	206.2 ± 4.4 (0.220)	−9. 89 ± 1.98	87 ± 1.1
5 mg in SLN	169.8 ± 4.7 (0.255)	−10.02 ± 2.66	78 ± 1.3
7 mg in SLN	185.9 ± 2.9 (0.265)	−10.20 ± 2.10	74 ± 1.8
9 mg in SLN	203.5 ± 5.9 (0.259)	−10.06 ± 2.66	75 ± 1.0
11 mg in SLN	215.3 ± 5.4 (0.222)	−18.96 ± 2.88	70 ± 2.1

**Table 6 nanomaterials-09-00230-t006:** Mean diameter ± standard error (PI) of empty nanoparticles suspensions in Pluronic^®^ F68 with sodium taurocholate (Na TC) as bile salt (SLN) and SLNs loaded with 9 mg CURC (SLN-CURC), before and after elution in GF and concentration with two different methods: concentration under N_2_ or freeze-drying.

	Mean Diameter (nm) ± S.E. (PI) Before Elution in GF	Mean Diameter (nm) ± S.E. (PI) After Elution in GF and Concentration under N_2_	Mean Diameter (nm) ± S.E. (PI) After Elution in GF and Concentration Freeze-Drying
SLN	152 ± 2.4 (0.21)	193 ± 0.6 (0.134)	302 ± 2.1 (0.198)
SLN-CURC	204 ± 5.9 (0.26)	173 ± 1.1 (0.123)	468 ± 1.9 (0.188)

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
