# Peer review of "Development of Solid Lipid Nanoparticles by Cold Dilution of Microemulsions: Curcumin Loading, Preliminary In Vitro Studies, and Biodistribution"

_nanomaterials, 2019, doi:10.3390/nano9020230_

Round 1

Reviewer 1 Report

My apologies for my negative response. I believe that the authors have published an excellent paper last year (Stearoyl-Chitosan Coated Nanoparticles Obtained by Microemulsion Cold Dilution Technique) describing application of cold emulsion method for preparing of CURC SNL. I can not recommend publication of this paper.

Authors can see my comments in the uploaded manuscript.

Author Response

Response to Reviewer 1

This manuscript regards the production of solid lipid nanoparticles by applying the microemulsion dilution technique recently developed in our laboratories. Contextually, chitosan coated nanoparticles were also prepared with the same technique and were object of a different manuscript. Even if CURC was chosen as model drug to be loaded in both nanoparticles series, the presence of a lipophilic ester of chitosan in nanoparticles composition should markedly modify in vivo fate. That is, the exploitation of a simple and versatile technique to produce lipid nanoparticles with different ingredients and having different physico-chemical and biological properties can be considered as an actual advantage rather than a duplication of products.

Reviewer 2 Report

The manuscript describes a novel method to formulate novel solid lipid nanoparticles containing curcumin. This reviewer thinks that the topic is novel and really interesting. The formulations are well characterised and tested. Accordingly, I think that it should be accepted for publication in this journal after some minor changes.

The phase diagrams are difficult to interpret. This reviewer thinks that indicating the different phases in the diagram will improve the readability significantly.

The scale bars in Figure 3 are blurry. Could the authors provide clarification about them in the figure caption?

The DSC thermograms should be incorporated to the manuscript. Is there a particular reason for omitting them?

In page 12, line 428 the authors claimed the curcumin concentration remained "quite unmodified during 1 month". This is not exactly true as there is as curcumin showed a clear tendency to decrease its concentration (30 ug/mL). This reviewer understand that the difference is really small (ca. 3%) but this sentence needs to be clarified.

Why are the curcumin release curves only up to 6h? The SLN release is around 10% only. It is obvious that the SLN provided a more sustained release but why did the authors stopped sampling at 6h?

In Figure 6 the cell viability study showed similar inhibition values for free curcumin and for the encapsulated one when the concentration was 10uM in PANC-1 cells. Why is this happening?

By having a look at Figure 7 this reviewer thinks that SLN provided a delayed curcumin release while increasing their stability. This can be easily seen as free curcumin provided a maximum accumulation in liver, lung and spleen at earlier stages than SLN (60 min vs 120 min). This fact should be commented  by the authors in the text. Moreover, the scales of the y axis in both panels of Figure 7 should be modified as it is difficult to compare the levels obtained for both formulations. An axis break in the first panel will be extremely beneficial and the second panel should show the y axis from 0 to 400 nmol CURC/kg of organ.

Finally, alternative works have been developed to develop sustained release systems for curcumin that could be cited in the present manuscript:

Larraneta E. et al; ACS Sustainable Chem. Eng.2018, 6, 9037-9046.

Larraneta E. et al; Int. J. Pharm. 2018, 538, 147-158.

Author Response

Response to Reviewer 2

The manuscript describes a novel method to formulate novel solid lipid nanoparticles containing curcumin. This reviewer thinks that the topic is novel and really interesting. The formulations are well characterized and tested. Accordingly, I think that it should be accepted for publication in this journal after some minor changes.

The phase diagrams are difficult to interpret. This reviewer thinks that indicating the different phases in the diagram will improve the readability significantly.

Phase diagrams have been re-drawn and the different existence area, previously omitted by a typographic misunderstanding, have been indicated.

The scale bars in Figure 3 are blurry. Could the authors provide clarification about them in the figure caption?

The scale bars have been removed from Figure 3 and included in the figure caption.

The DSC thermograms should be incorporated to the manuscript. Is there a particular reason for omitting them?

DSC thermograms, not previously included in order to avoid making the manuscript too cumbersome, have been incorporated as Supplementary Material.

In page 12, line 428 the authors claimed the curcumin concentration remained "quite unmodified during 1 month". This is not exactly true as there is as curcumin showed a clear tendency to decrease its concentration (30 ug/mL). This reviewer understand that the difference is really small (ca. 3%) but this sentence needs to be clarified.

We agree with the Reviewer’s opinion. Although the variation over time was very low, looking at the graph it seemed not to be: for this reason, the y axis scale has been opportunely reduced.

Why are the curcumin release curves only up to 6h? The SLN release is around 10% only. It is obvious that the SLN provided a more sustained release but why did the authors stopped sampling at 6h?

Curcumin released was followed up to 24 h and, even if not reported in the graph, curcumin released amounts both from SLN and from the reference solution have been introduced in the text.

In Figure 6 the cell viability study showed similar inhibition values for free curcumin and for the encapsulated one when the concentration was 10uM in PANC-1 cells. Why is this happening?

It might be, and our previous experience seems to support this assumption, that PANC-1 cell can internalize curcumin in greater extent than CFPAC-1 ones.

By having a look at Figure 7 this reviewer thinks that SLN provided a delayed curcumin release while increasing their stability. This can be easily seen as free curcumin provided a maximum accumulation in liver, lung and spleen at earlier stages than SLN (60 min vs 120 min). This fact should be commented by the authors in the text. Moreover, the scales of the y axis in both panels of Figure 7 should be modified as it is difficult to compare the levels obtained for both formulations. An axis break in the first panel will be extremely beneficial and the second panel should show the y axis from 0 to 400 nmol CURC/kg of organ.

In Figure 7 (B) y axis was changed according to Reviewer’s suggestion from 0 to 400 nmol/kg of organ. Authors have commented the different accumulation values in liver, lung and spleen of CURC when released from SLN compared to solution at different time.

Finally, alternative works have been developed to develop sustained release systems for curcumin that could be cited in the present manuscript:

- Larraneta E. et al; ACS Sustainable Chem. Eng.2018, 6, 9037-9046.

- Larraneta E. et al; Int. J. Pharm. 2018, 538, 147-158.

We cited the latter article in the References

Reviewer 3 Report

The study deals with a new easy to prepare slow releasing dug delivery system for curcumin parenteral application. The experimental design and performance are appropriate, well described and precisely conducted.

There are some minor changes to be made:

Please indicate the source of human immortalized podocytes.

Please indicate the constituents of the complete medium and keeping in growth procedure for the pancreatic cancer cell lines CFPAC-1 and PANC-1.

The studied SLN curcumin formulation has a different from the free curcumin tissue accumulation profile (in spleen, lung and liver predominantly) and this fact needs to be underlined and discussed in details. Noteworthy, the SLN curcumin formulation does not reach higher concentrations in the pancreatic tissue of the experimental animals. Is this a disadvantage or not?

Author Response

Response to Reviewer 3

The study deals with a new easy to prepare slow releasing drug delivery system for curcumin parenteral application. The experimental design and performance are appropriate, well described and precisely conducted.

There are some minor changes to be made:

Please indicate the source of human immortalized podocytes.

The source has been indicated in the text

Please indicate the constituents of the complete medium and keeping in growth procedure for the pancreatic cancer cell lines CFPAC-1 and PANC-1.

Medium and procedure for cell growth were indicated in the text.

The studied SLN curcumin formulation has a different from the free curcumin tissue accumulation profile (in spleen, lung and liver predominantly) and this fact needs to be underlined and discussed in details. Noteworthy, the SLN curcumin formulation does not reach higher concentrations in the pancreatic tissue of the experimental animals. Is this a disadvantage or not?

The discussion of in vivo data has been improved. Figure 7 has been modified and a zoom of pancreas section has been introduced, to evidence that at each post-administration time CURC pancreatic accumulation is always higher from SLN than from the reference solution.

Round 2

Reviewer 1 Report

Authors have not addressed my concerns or provided any explanation. May be, I do not have sufficient expertise with SLNs to review this manuscript. Other reviewers had more positive opinion, thus the Editors can decide the fate of this manuscript based on the comments from other reviewers.

Reviewer 2 Report

The authors have addressed all the reviewer's comments. This reviewer thinks that the manuscript should be accepted for publication.

Reviewer 3 Report

The manuscript was correctly revised.